# Effect of Grain-Size in Nanocrystalline Tungsten on Hardness and Dislocation Density: A Molecular Dynamics Study

Toufik Karafi [1,*], Abdellah Tahiri [2], Hanae Chabba [3], Mohamed Idiri [1] and Brahim Boubeker [1]

1   LIMAT Laboratory, University Hassan II, Faculty of Sciences Ben M'Sik, Casablanca B.P. 7955, Morocco
2   LPAIS Laboratory, University Sidi Mohamed Ben Abdellah, Faculty of Sciences, B.P. 1796, Fez-Atlas 30003, Morocco
3   LPEDD Laboratory, Superior School of Technology Fez, University Sidi Mohamed Ben Abdellah, Fez B.P. 1796, Morocco
*   Correspondence: karafi.toufik9@gmail.com; Tel.: +212-610-85-74-92

**Abstract:** We have simulated a series of nanoindentation experiments on nanocrystalline tungsten specimens using a combination of molecular dynamics simulations and the embedded atom method potential. The research aimed to investigate the impact of grain size on the mechanical properties of tungsten. Nanoindentation is a technique used to measure the mechanical properties of materials at a small scale. In this study, the researchers varied the grain size of the tungsten specimens, ranging from 7.9 to 10.5 nanometers. They also applied a loading rate of 3 angstroms per picosecond at a temperature of 300 Kelvin. The study found that as the grain size increased, the hardness increased, and the elastic modulus decreased. Hardness is a measure of a material's resistance to deformation, and the elastic modulus is a measure of a material's stiffness. The findings suggest that as the grain size of tungsten increases, the material becomes harder but less stiff. Additionally, the study explored the ways in which nanocrystalline tungsten deformed during nanoindentation. The researchers found that the deformation of the material was primarily due to dislocation activity, which is consistent with previous research on the topic. Overall, the findings of this research provide valuable insights into the mechanical properties of nanocrystalline tungsten and the ways in which the material deforms under stress. These findings could have practical applications in the development of materials for use in various industries.

**Keywords:** nanocrystalline tungsten; nanoindentation; molecular dynamics simulations; mechanical properties; defect mechanisms; hardness

## 1. Introduction

Nanocrystalline materials are a class of materials that have grain sizes on the order of nanometers, typically ranging from 1 to 100 nm. Compared to their coarse-grained counterparts, which have grain sizes on the order of micrometers, nanocrystalline materials exhibit unique mechanical properties, such as enhanced strength, hardness, toughness, and diffusivity [1].

The enhanced strength of nanocrystalline materials can be attributed to a combination of factors, including grain size refinement, high dislocation density, and grain boundary strengthening. In nanocrystalline materials, the small grain size means that there are more grain boundaries per unit volume, which impedes the movement of dislocations and reduces plastic deformation. This results in an increase in strength, as well as a decrease in ductility [2].

In addition to increased strength, nanocrystalline materials also exhibit enhanced hardness and toughness. The high density of grain boundaries in nanocrystalline materials creates a large number of obstacles for dislocation movement, which increases the material's resistance to deformation and improves its hardness. Meanwhile, the high density of defects in nanocrystalline materials, such as grain boundaries, dislocations, and vacancies, can

increase the material's toughness by providing additional sources of energy dissipation during deformation [3].

Finally, the enhanced diffusivity of nanocrystalline materials is due to the large number of grain boundaries and interfaces, which act as pathways for mass transport. This can have important implications for applications such as catalysis and hydrogen storage.

Overall, the unique properties of nanocrystalline materials have made them an area of active research in materials science over the past decade. Researchers have explored a variety of techniques for synthesizing nanocrystalline materials, including mechanical milling, severe plastic deformation, and electrodeposition, among others. However, despite significant progress, challenges remain in terms of characterizing the mechanical properties of nanocrystalline materials and understanding the fundamental mechanisms that govern their behavior.

Nanocrystalline tungsten (nc-W) has attracted significant attention in recent years due to its unique mechanical properties, which are significantly different from those of conventional tungsten. The small grain size of nc-W leads to a high density of grain boundaries, which can enhance its strength, ductility, and fracture toughness. Moreover, nc-W exhibits good resistance to radiation damage, making it a potential candidate for fusion and nuclear applications. Several studies have investigated the synthesis, characterization, and mechanical properties of nc-W using different methods, such as mechanical alloying, severe plastic deformation, and thermal processing. For example, Li et al. [4] studied the effect of grain size and temperature on the mechanical behavior of nc-W using in situ transmission electron microscopy. They found that the mechanical properties of nc-W were strongly dependent on the grain size and temperature. Similarly, Zhang et al. [5] investigated the microstructure and mechanical properties of nc-W fabricated by spark plasma sintering. They found that nc-W exhibited high strength and ductility due to its fine-grained microstructure. These studies demonstrate the potential of nc-W for various applications and highlight the importance of understanding its unique mechanical properties.

The Hall–Petch (H–P) relation has been used to verify that the strength and hardness of nanocrystalline materials increase with decreasing grain size down to a critical size of 10–20 nm [6]. Of particular interest is nanocrystalline tungsten (W) due to its high melting point, excellent high-temperature mechanical properties, and resistance to sputtering, which make it suitable for use in nuclear fusion components, microelectronics, welding electrodes, and high-voltage electrical contacts [7–9].

Nanoindentation testing is a simple yet powerful technique for evaluating the mechanical properties of materials, such as hardness, elastic modulus, and defect mechanisms, and provides insight into various phenomena, including cracking mechanisms, fracture toughness, strain-hardening, creep, and defect nucleation [10–17]. The combination of nanoindentation testing with modern experimental methods and Oliver–Pharr analysis has led to its widespread use [18].

The EAM potential [19] is a type of many-body potential that takes into account the effect of the electron density on the interatomic interactions. It was originally developed for studying metallic systems, but it has since been extended to other materials, such as semiconductors and ceramics. The EAM potential can accurately capture a wide range of properties, including crystal structures, melting temperatures, and surface properties.

One advantage of MD [20] simulations using EAM potentials is that they can provide a detailed understanding of the structure and dynamics of materials at the atomic and molecular level. For example, they can be used to study the formation and behavior of defects, such as dislocations and vacancies, which play a crucial role in determining the mechanical properties of materials. They can also be used to study the behavior of materials under different conditions, such as high pressure or temperature.

Another important application of MD [21] simulations using EAM potentials is in the design and optimization of materials for specific applications. By simulating the properties of different materials, researchers can identify materials with desirable properties and optimize their structures to enhance their performance. For example, MD simulations

can be used to design materials with high strength, low thermal expansion, or improved catalytic activity.

In recent years, there has been increasing interest in using MD [22] simulations with machine learning (ML) techniques to accelerate materials discovery and design. ML algorithms can be trained on large datasets of MD simulations to predict the properties of new materials without the need for expensive simulations. This approach, known as "materials informatics", has the potential to significantly accelerate the development of new materials for a wide range of applications.

In this study, we used MD simulations to investigate the effect of grain size on the mechanical behavior and structural properties of nanocrystalline tungsten (W) through a series of nanoindentation tests on samples with grain sizes ranging from 7.9 nm to 10.5 nm. The results showed that the reduced elastic modulus, hardness, and yield stress of nanocrystalline tungsten decreased with increasing grain size, in line with the H–P relation. Additionally, the number of dislocations increased with increasing indentation depth during loading, and the dislocation density increased with decreasing grain size, The study was able to identify different types of dislocation structures based on their Burgers vectors, such as those with b = 1/2 <111>, b = <100>, and b = <110>. Previous studies on molybdenum, iron, tantalum, and vanadium have also shown that the slip of edge dislocations in body-centered cubic (bcc) metals is faster than the slip of screw dislocations, and samples with finer grains tend to have improved ductility due to the increased density of slip dislocations.

## 2. Materials and Methods

### 2.1. Interatomic Potential

Molecular dynamics (MD) simulations are widely used to investigate the behavior of materials at the atomic level. In MD simulations, the choice of interatomic potential function is crucial as it describes the energy between two atoms in the system. The embedded atomic method (EAM) potential is one such function that was initially developed by Daw and Baskes in 1984 to study defects in metals [23]. The EAM potential calculates the energy between atoms as the sum of functions that consider the separation between atoms and their neighbors. The total energy of a system with N atoms, represented by $E_{tot}$, is given by Equation (1):

$$E_{tot} = \sum_i F_i(\rho_i) + \frac{1}{2} \sum_{i \neq j} \varphi_{i,j}(r_{i,j}), \rho_i =_{j \neq i} f_i(r_{ij}) \tag{1}$$

where $E_{tot}$ is the total energy of the system, $\varphi_{i,j}(r_{i,j})$ represents the pair interaction energy between an atom i and its neighboring atom j, $f_i(r_{ij})$ is the electronic density function, and $F_i(\rho_i)$ represents an embedding function accounting for the effects of the free electrons in the metal [24].

### 2.2. Sample Preparation

The molecular dynamics (MD) simulations in this study were performed using the open-source code LAMMPS [25] and the embedded atomic method (EAM) potential developed by Bonny and Bakaev [24]. The simulation model consisted of a spherical diamond indenter with a radius of 30 Å and a simulation box of nanocrystalline tungsten with various grain sizes and a cubic body center structure (BCC). The simulation box contained around 504,948 atoms with dimensions of 200 Å × 200 Å × 200 Å in the x, y, and z directions. The simulation employed periodic boundary conditions in the x and y directions and a free surface in the z direction, with a time step of 1fs.

To construct the simulation box, three types of atoms were used: fixed atoms, thermostat atoms, and Newtonian atoms. The substrate's lower layer of atoms was kept fixed in space, while the adjacent layer was maintained at a constant temperature of 300 K. The indenter velocity was set to 3 Å/ps. Table 1 provides an overview of the simulation parameters used in this study, and Figure 1 illustrates the nanoindentation process.

**Table 1.** Computational details used for the development of the MD simulation model.

| Material | W |
|---|---|
| Number of atoms | ~504,948 |
| Box size | 200 (x) × 200 (y) × 200 (z) Å |
| Time-step (fs) | 1 |
| Temperature (K) | 300 |
| Potential | EAM |
| Pressure (Pa) | Ambient atmospheric |
| Ensemble | NVT |
| Boundary Conditions | PPP |
| Runs | 30,000 |
| Indenter velocity | 3 Å/ps |
| Indenter radius | 30 Å |

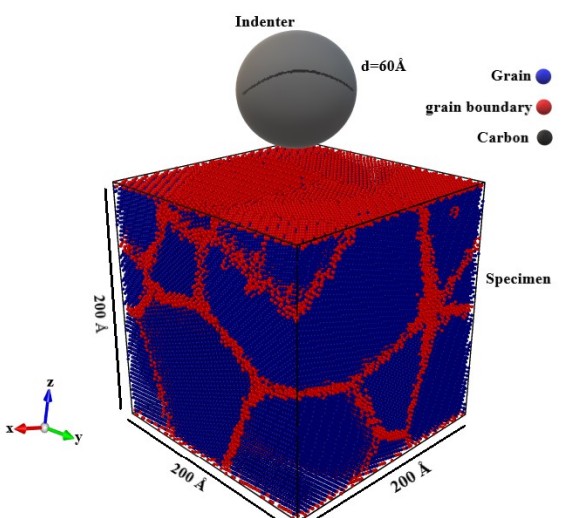

**Figure 1.** A simulation model can be created to illustrate the nanoindentation process in three dimensions using a spherical indenter as a representative example, where the particles colored blue, red, and black represent grain, grain boundary, and carbon atoms, respectively.

To investigate the mechanisms of dislocation during the nanoindentation process, researchers have employed the OVITO software [26] in combination with the centrosymmetry parameter (CSP) [27], common neighbor analysis (CNA) [27], and dislocation extraction algorithm (DXA) [27] to analyze the results of their molecular dynamics (MD) simulations. These tools have been widely used to study the atomic-level structure and behavior of materials under different mechanical and environmental conditions.

The OVITO software provides powerful visualization and analysis capabilities for large-scale atomistic systems, making it an ideal tool for studying the behavior of materials during simulations. The CSP is used to determine the degree of centrosymmetry in the local atomic environment, while the CNA is used to identify and characterize crystal defects, such as grain boundaries, stacking faults, and dislocations. The DXA is used to extract and analyze dislocations from the MD simulations, providing valuable insights into the mechanisms of dislocation during the nanoindentation process.

By employing these analysis tools in their studies, researchers can gain a better understanding of the behavior of materials under different conditions, which can ultimately lead to the development of new and improved materials for various applications. Therefore, the use of these tools and techniques can significantly contribute to the advancement of materials science and engineering.

## 3. Results and Discussions

### 3.1. Effect of Size-Grain on Hardness and Elastic Modulus

To calculate the Hertzian force, the average of the instantaneous indenter force over the last 100 femtoseconds prior to the downward movement of the indenter was taken. The expression for the Hertzian force is as follows:

$$F = \frac{4}{3}\left(K.h^{\frac{3}{2}}\right); K = E_r.R^{1/2} \tag{2}$$

To obtain the indenter force, the instantaneous force was averaged over the last 100 femtoseconds before moving the indenter downwards. The hardness value, denoted by *H*, is then calculated using the following expression:

$$H = \frac{F_{max}}{A} \tag{3}$$

where $F_{max}$ is the maximum load and *A* is the projected contact area, calculated using the following equation [28]:

$$A \approx 2\pi R h_{res} \tag{4}$$

where *R* is the radius of the indenter and $h_{res}$ is the residual depth.

We determine the dislocation density ($\rho_{disl}$) by [29]:

$$\rho_{disl} = \frac{Total\,length\,of\,dislocation}{2\pi R_{plast}^3/3 - 2\pi R^3/3} \tag{5}$$

where $R_{plast}$ is the radius of the plastic zone and *R* is maximum indentation to depth.

The nanoindentation process involves the use of different grain sizes, namely 7.9 nm, 8.5 nm, 9.1 nm, and 10.5 nm. The dimensions of the box are kept constant at 200 Å × 200 Å × 200 Å as previously mentioned. In addition, the loading rate is fixed at 3 Å/ps, and the temperature is maintained at 300 K for each grain size.

The passage describes Figure 2, which shows load and unload displacement curves for various grain sizes during indentation and retraction. The rigid indenter moved 6 Å to reach the specimen's surface, resulting in a total displacement of 39 Å with a velocity of 3 Å/ps. The contact point between the indenter and the specimen depends on the substrate's ability to adjust and relax under the interactions with the tip, which is crucial in determining the contact location.

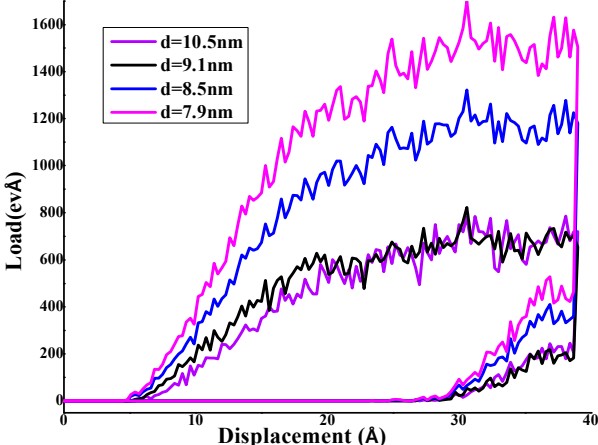

**Figure 2.** Load and unload displacement (P-h) line graphs for the nanocrystalline tungsten (W) uder a spherical nanoindentation along z at 300K and 3 Å/ps, the average grain size ranging from: 7.9 (color purple), 8.5 (color blue), 9.1(color black) and 10.5 (color magenta).

Indentation and retraction experiments are commonly used in materials science to study the mechanical properties of materials at small length scales. In this case, the experiments were performed on specimens with different grain sizes, and the resulting curves provide information about the deformation behavior and elastic properties of the materials.

The movement of the indenter and the resulting displacement values are important parameters to consider in the interpretation of the experimental data. Additionally, the dependence of the contact location on substrate relaxation underscores the importance of careful sample preparation and handling in these experiments. Overall, the information presented in the passage provides insight into the experimental setup and considerations involved in the indentation and retraction process.

The passage mentions Table 2, which lists the calculated results of the reduced elastic modulus, yield stress, and hardness for four different sizes of grains. These properties are important measures of a material's mechanical behavior and can provide insights into its deformation mechanisms and strength.

**Table 2.** Hardness measurements in GPa, for all concentrations.

| Size of Grain (nm) | d = 7.9 | d = 8.5 | d = 9.1 | d = 10.5 |
|---|---|---|---|---|
| Reduced elastic modulus $E_r$ (GPa) | 497.25 | 424.12 | 292.5 | 235.2 |
| Hardness $H$ (GPa) | 54.7 | 42.58 | 26.51 | 24.73 |
| Yield stress $\sigma_e$ (GPa) | 57.2 | 43.89 | 28.2 | 24.07 |

The reduced elastic modulus is a measure of a material's stiffness and is calculated as the ratio of the applied stress to the resulting strain. It is often used to compare the mechanical properties of materials with different sizes or geometries. The yield stress is the stress at which a material begins to exhibit plastic deformation, or a permanent change in shape, and is an important measure of its strength. The hardness is a measure of a material's resistance to deformation or indentation and is often used to compare the wear resistance of different materials.

In Table 2, the values of $E_r$ (reduced elastic modulus) and $\sigma_e$ (yield stress) were found to vary considerably, indicating that the mechanical properties of the material change with different grain sizes. Specifically, as the grain size increased, the reduced elastic modulus and yield stress decreased, which can be attributed to the increased occurrence of grain boundaries in larger grain sizes.

To further explore the effect of grain size on mechanical properties, Zhechen employed the potential embedded H and He to calculate the Young's modulus of nanocrystalline W and obtained a value of 311.7 GPa for an average grain size of 5 nm [30]. This value can be used as a reference for comparison with the results obtained from the different grain sizes.

To compare the different grain sizes, the Oliver and Pharr method [18] was utilized to determine the reduced elastic modulus (Er) for an indentation depth of 15 Å. This method is widely used in nanoindentation experiments to measure mechanical properties such as hardness and reduced modulus. The indentation depth of 15 Å was chosen as it falls within the range of shallow indentations, which can provide accurate measurements of the mechanical properties of the material.

The study of nanocrystalline metals and their mechanical properties has been an area of intense research in recent years. The hardness of nanocrystalline metals is known to be strongly influenced by grain size, with smaller grain sizes resulting in higher hardness. This phenomenon is known as the Hall–Petch effect, which has been observed in a wide range of nanocrystalline metals, including tungsten, tantalum, copper, and aluminum [31–33].

In the case of tungsten, the hardness values obtained from the simulations in this study are consistent with previous experimental and simulation studies. For example, Saurav Goel reported a hardness value of 42 GPa for single crystal tungsten using a rigid diamond indenter and an embedded atom method (EAM) potential [34]. In comparison,

the hardness values obtained in this study for tungsten nanocrystals range from 24.73 to 54.7 GPa, with lower values associated with larger grain sizes.

The relationship between grain size and hardness in nanocrystalline metals is typically described by the Hall–Petch equation, which states that the hardness is inversely proportional to the square root of the grain size. The Hall–Petch equation is expressed as $H = H_0 + \frac{K}{\sqrt{d}}$, where H is the hardness, H0 is the intercept of the equation, k is the slope, and d is the grain size [35]. Figure 3 in the original text of this study illustrates this relationship for the tungsten nanocrystals simulated in this study.

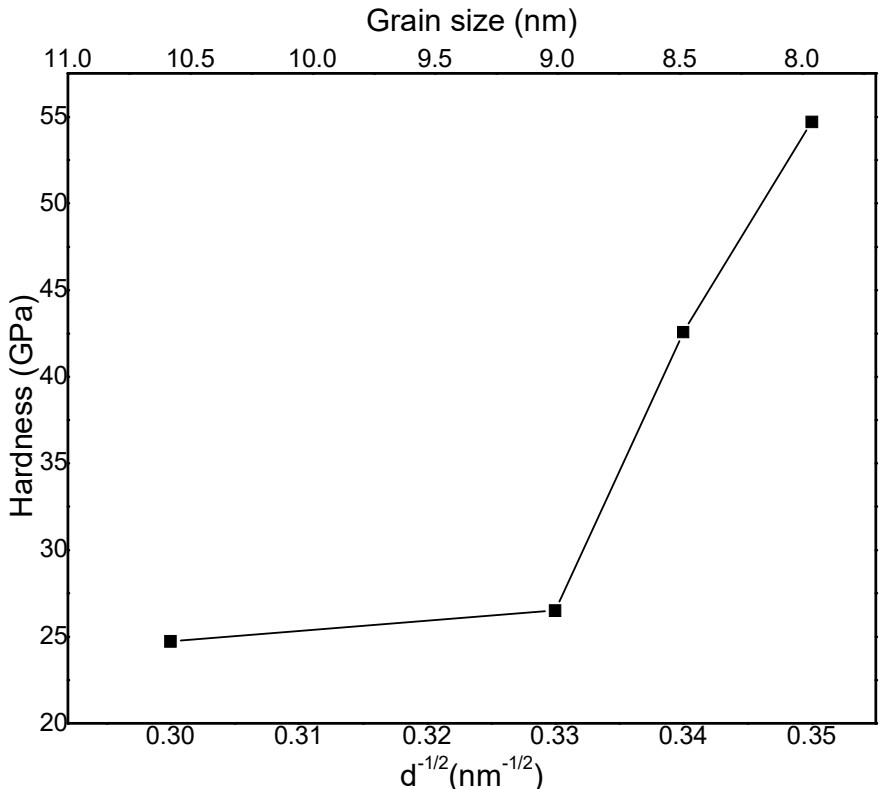

**Figure 3.** Size of grain vs. hardness in this MD study: Hall–Petch (HP) relationship for the nanocrystalline W cases.

In addition to the Hall–Petch effect, other factors can also influence the mechanical properties of nanocrystalline metals, such as grain boundary effects, dislocation density, and residual stress. The understanding of these factors is crucial for the development of nanocrystalline metals with tailored mechanical properties for various applications.

### 3.2. Mechanisms of Dislocation during Nanoindetation Process

In this section, we delve into a molecular dynamics (MD) simulation study that aimed to compare the mechanisms of dislocation during nanoindentation in pure tungsten (W) with varying grain sizes ranging from 7.9 nm to 10.5 nm.

Figure 4 describes the findings of a study that used molecular dynamics (MD) simulations and various analysis tools to investigate the mechanisms of dislocation during the nanoindentation process in tungsten. The study was able to identify different types of dislocation structures based on their Burgers vectors, such as those with b = 1/2 $\langle 111 \rangle$, b = $\langle 100 \rangle$, and b = <110>, among others, as shown in Figure 4 of the study. These dislocation structures were found to occur at an indentation depth of 39 Å.

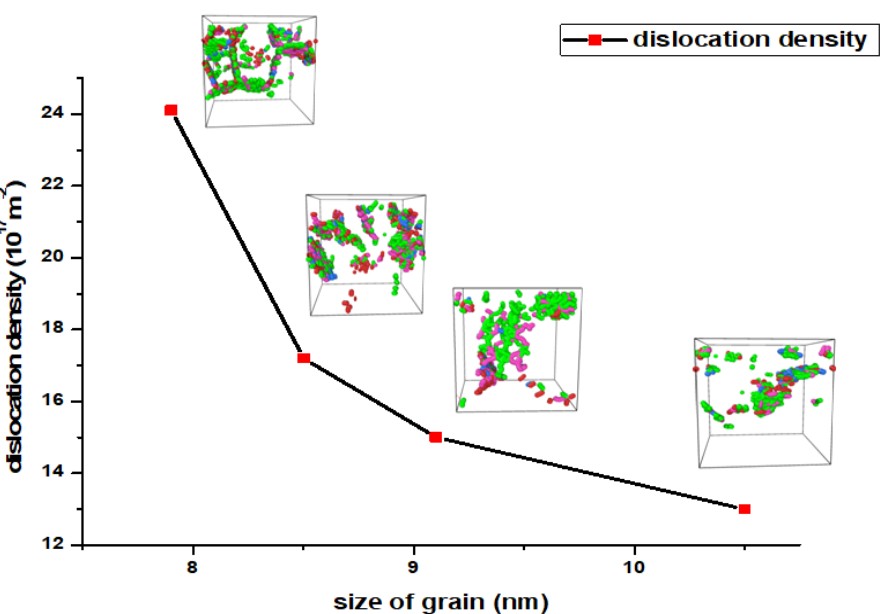

**Figure 4.** Dislocation density vs. grain size with the dislocation analysis results of a nanodentation simulation showing the extracted dislocation line network for varied grain sizes.

As the indentation depth increased, the number of dislocations and their density also increased. The study found that the dislocation density was inversely proportional to the grain size of the tungsten sample. This finding is consistent with previous research on molybdenum, which is a material similar to tungsten. In a previous study, it was found that a decrease in grain size led to an increase in the density of edge/mixed dislocations in molybdenum [36].

Furthermore, the text notes that previous studies on other body-centered cubic (bcc) metals, such as iron, tantalum, and vanadium, have shown that the slip of edge dislocations is faster than the slip of screw dislocations [37–39]. This information may be relevant in predicting the behavior of tungsten and other BCC metals under stress and strain conditions.

## 4. Conclusions

The aim of the present study was to investigate the impact of grain size on the mechanical properties of nanocrystalline tungsten (W) during nanoindentation deformation, using molecular dynamics (MD) simulation. The study focused on examining the mechanisms of dislocation in nanocrystalline tungsten (W) and their effects on the material's mechanical behavior.

The study's findings suggest that as the grain size of tungsten increased, the reduced elastic modulus, hardness, and yield stress of the material decreased. These results are consistent with the conventional Hall–Petch tendency, which proposes that smaller grain sizes lead to higher hardness. The study's results indicate that the mechanical properties of nanocrystalline tungsten (W) are significantly influenced by its grain size.

The study also found that during loading, the number of dislocations in the material increased with increasing indentation depth. Moreover, the dislocation density increased with decreasing grain size. These results suggest that dislocation activity plays a critical role in the deformation of nanocrystalline tungsten (W) during nanoindentation.

Overall, the study's findings provide valuable insights into the mechanical properties and dislocation mechanisms of nanocrystalline tungsten (W) during nanoindentation. The results could be useful in developing materials for various industries, such as aerospace, automotive, and energy.

**Author Contributions:** Conceptualization, methodology, and software, T.K.; validation, A.T.; formal analysis, investigation, and resources, M.I.; data curation, writing—original draft preparation, H.C.; writing—review and editing, T.K.; visualization, supervision, project administration, and funding acquisition, B.B. All authors have read and agreed to the published version of the manuscript.

**Funding:** This research received no external funding.

**Data Availability Statement:** Data are contained within the article.

**Acknowledgments:** Although they may not agree with all interpretations or conclusions of this work, we thank our colleagues at Sidi Mohamed Ben Abdellah University for their insight and skills, which greatly aided the research.

**Conflicts of Interest:** The authors declare no conflict of interest.

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
