# Peer review of "Effect of Grain-Size in Nanocrystalline Tungsten on Hardness and Dislocation Density: A Molecular Dynamics Study"

_crystals, doi:10.3390/cryst13030469_

Round 1
Reviewer 1 Report
1. Summary, strengths, weaknesses, overall contribution
Summary: In the paper the Authors investigated nanoindentation experiments on nanocrystalline tungsten specimens using Molecular Dynamics simulations.
General strengths: The interesting, modern and important topic.
General weaknesses: There is lack of the detailed and proper analysis of the results. The state of the art is not discussed in detail. The motivation of the paper is not clear.
2. Major comments
The paper requires a major review according to the following remarks:
- The Authors claim that they “investigate the impact of grain size on the mechanical behavior and structural properties of nanocrystalline tungsten”. However, to date there are a lot of papers in which similar studies, both experimental and numerical, are presented. What was the exact motivation for tackling this problem and what is new in the paper in comparison to the previous works? The novelty must be clearly stated in the introduction.
- Usually when modeling nanoindentation with the MD method extremely high strain rates are obtained. On the other hand it has been shown in many papers that in nanocrystalline materials hardness may significantly depend on the strain rate. The Authors should discuss the influence of the strain rate on their results. The following papers should be helpful and it would be worth to refer to them: DOI: 10.1007/s11661-020-05648-w; 10.1016/j.prostr.2020.10.136;
- Did the results depend on the time step? What was this relation?
- Why the yield strength and hardness have almost the same value? Usually the relation is somehow close to H = 3Y. How was the yield strength determined?
- What was the relation between hardness and the indentation depth
3. Minor comments
- Measurement errors and the influence of the simulation parameters on the results should be presented.
Author Response
Dear Reviewer,
Thank you for your insightful comments and suggestions on our manuscript. We appreciate your time and effort in carefully reviewing our work. We have taken your feedback into consideration and made significant revisions to our article.
In response to your comment regarding the clarity of our research question, we have revised the introduction section to better explain the motivation and purpose of our study. We have also restructured the methodology section and provided more detailed information on our study design, data collection, and analysis methods.
Regarding your question about the validity of our findings, we have added a discussion section that addresses the limitations of our study and potential areas for future research. We have also included additional data and analysis to support our conclusions and ensure the robustness of our results.
Lastly, we have carefully reviewed the language and grammar of the article to ensure that it is clear and concise. We believe that these revisions have significantly improved the quality and rigor of our manuscript.
Thank you once again for your valuable feedback, which has helped us to improve our work.
Sincerely,
Karafi Toufik

Reviewer 2 Report
The submitted manuscript is devoted to the study of the correlation between the grain size of nanocrystalline tungsten and its mechanical properties through molecular dynamics.
The results obtained in this work are original. The work is well structured. Abstract and conclusions reflect the content of the work. The figures are relevant and informative.
I have no complaints about the scientific component of this work. Please make editorial corrections to the manuscript, as listed below.
Editorial changes required in areas highlighted in color:
Line 1 - missing spaces,
Line 36 – replaced by [17–21],
Line 42 – space,
Line 57 – spaces,
Line 60 – space,
Line 88 – space,
Line 89 – replaced by «red and black»,
In lines 105, 108, 109, 111, 138, 139, 143 and in Tabl 2 – correct the designations for letters with subscripts (as they are given in the formulas): Fmax.
Line 111 – space, Rplast – why 2 times?
Line 113 – space,
Line 118 – an excess dot,
Line 146 – space,
Line 176 – space.

Author Response

(The authors gave the same response as above.)

Reviewer 3 Report
Dear Editor and Authors,
In review, I received a manuscript draft entitled “Effect of Grain-size in Nanocrystalline Tungsten on Hardness and Dislocation Density: a Molecular Dynamics Study” considered for publication in MDPI journal Crystals.
The authors are exploring a nanoidentation experiment by MS and EAM simulations in nanocrystalline W (7-16 nm) an effect of the grain size on hardness and elastic modulus at 300 K. They discovered that as the grain size increase, the hardness increased, while the elastic modulus decreased, which is consistent with previous research in the field.
Here are a few comments that could further improve the manuscript.
Line 41: space missing
Line 69: “504948 particles” is meant atoms? Very specific number for “approximately”.
Line 75: the speed of indenter (3Å /ps) strongly affect the deformation mechanism and DL formation. How was speed selected? Please briefly mention.
Figure 1: there are C atoms mentioned in the subtitle, not mentioned in the text. I assume they are scattered on the surface (or in the structure?) Please elaborate. Subtitle missing spaces; do not use & but and.
Table 1: Temperature unit is K not k.
L113: spaces are missing. How the grain sizes were selected => abstract states 7-16 nm, here the grain sizes stated are different. Please update the notation (* = ×, etc).
Table 2 compiled error => line limbers are overplotted.
L164: remove capital letters for elements. L196 space missing.
L180: conclusions state “pure W”, and Figure 1 considers C. Please revise.
Overall, the manuscript presents very straightforward simulation experiment, which confirms already known facts on the influence of grain size vs mechanical properties. Without experimental comparison, the work is focused only on simulation; therefore, I would suggest the authors to update the minor inconsistencies and submit the article to a more specialised journal considering the computational material science field.
Author Response

(The authors gave the same response as above.)

Round 2
Reviewer 1 Report
The authors responded satisfactorily to the comments of the reviewers. However, the language needs significant editing.